# Changes in the Ultrastructure of *Staphylococcus aureus* Cells Make It Possible to Identify and Analyze the Injuring Effects of Ciprofloxacin, Polycationic Amphiphile and Their Hybrid

**DOI:** 10.3390/microorganisms11092192

**Published:** 2023-08-30

**Authors:** Alina E. Grigor’eva, Alevtina V. Bardasheva, Elena S. Ryabova, Anastasiya V. Tupitsyna, Danila A. Zadvornykh, Lyudmila S. Koroleva, Vladimir N. Silnikov, Nina V. Tikunova, Elena I. Ryabchikova

**Affiliations:** Institute of Chemical Biology and Fundamental Medicine, Siberian Branch of Russian Academy of Science, Lavrent’ev av., 8, 630090 Novosibirsk, Russia; feabelit@mail.ru (A.E.G.); herba12@mail.ru (A.V.B.); e.ryabova@g.nsu.ru (E.S.R.); aysa@ngs.ru (A.V.T.); d.zadvornykh@alumni.nsu.ru (D.A.Z.); koroleva@niboch.nsc.ru (L.S.K.); silnik@niboch.nsc.ru (V.N.S.); tikunova@niboch.nsc.ru (N.V.T.)

**Keywords:** transmission electron microscopy, *Staphylococcus aureus*, planktonic *S. aureus*, polycationic amphiphile, ciprofloxacin, hybrid antibiotic, light microscopy, *S. aureus* biofilm, biofilm matrix

## Abstract

The purposeful development of synthetic antibacterial compounds requires an understanding of the relationship between effects of compounds and their chemical structure. This knowledge can be obtained by studying changes in bacteria ultrastructure under the action of antibacterial compounds of a certain chemical structure. Our study was aimed at examination of ultrastructural changes in *S. aureus* cells caused by polycationic amphiphile based on 1,4‒diazabicyclo[2.2.2]octane (DL_4_12), ciprofloxacin and their hybrid (DL_5_Cip6); the samples were incubated for 15 and 45 min. DL_4_12 first directly interacted with bacterial cell wall, damaging it, then penetrated into the cell and disrupted cytoplasm. Ciprofloxacin penetrated into cell without visually damaging the cell wall, but altered the cell membrane and cytoplasm, and inhibited the division of bacteria. The ultrastructural characteristics of *S. aureus* cells damaged by the hybrid clearly differed from those under ciprofloxacin or DL_4_12 action. Signs associated with ciprofloxacin predominated in cell damage patterns from the hybrid. We studied the effect of ciprofloxacin, DL_4_12 and their hybrid on *S. aureus* biofilm morphology using paraffin sections. Clear differences in compound effects on *S. aureus* biofilm (45 min incubation) were observed. The results obtained allow us to recommend this simple and cheap approach for the initial assessment of antibiofilm properties of synthesized compounds.

## 1. Introduction

The number of deaths in the world associated with bacterial infections in 2019 amounted to 7.7 million, and more than half of them were caused by five pathogens, including 1 million deaths due to *Staphylococcus aureus*, once again indicating the harmfulness of this bacterium [1].

The acquisition of multidrug resistance by pathogenic bacteria, and the severe consequences of this phenomenon that threaten humanity, have induced active research aimed at developing new antimicrobial agents capable of combating antibiotic-resistant species [2,3]. Hybrid antibiotics, which do not exist in nature, are a type of such agents, and various strategies for assembling hybrids from different compounds have been published, including antibiotics known and used in practice. A recent review by Lungu and coauthors [2] provided a comprehensive analysis of hybrid antibiotics based on fluoroquinolones, and their composition and activity. Fluoroquinolones are a group of antibiotics with initially high antibacterial efficacy, but now, the “major” pathogens have developed resistance to many of them. The chemical and biological properties of fluoroquinolones determine their promise as components of hybrids; their hybrids with other antibiotics and active compounds have been described, and some of them are under clinical trials [2,4].

Much attention of researchers is paid to ciprofloxacin (fluoroquinolone of the second generation), which is highly effective in infectious diseases; this antibiotic allows its use in various dosage forms and is well tolerated by patients [2]. In addition to hybrids of ciprofloxacin with other antibiotics, hybrids with non-antibiotics have been synthesized and are being studied; non-antibiotics are represented by compounds of different chemical structure [2,5].

Polycationic amphiphiles based on 1,4‒diazabicyclo[2.2.2]octane (DABCO) have a wide spectrum of biological activity: antibacterial, antifungal and antiviral [6,7,8]. Structurally, these compounds consist of two DABCO residues with a long lateral hydrophobic substituent connected by a linker group. In this work, we used a compound called “DL_4_12” (Figure 1), in which quaternary residues of DABCO (D) were connected by tetramethylene linker (L_4_) with a dodecane substituent (12). The synthesis of DL_4_12 and data on its antibacterial properties were published in [6]. This compound showed high effectiveness against five Gram-negative and eight Gram-positive bacteria (including *S. aureus* ATCC 25923 and methicillin-resistant *S. aureus* strains CEMTC 675, 707, 1695, 1732, 1733 and 1850). The ciprofloxacin, used as a standard, showed lower activity than DL_4_12 against the same bacteria [6].

Polycationic antimicrobial compounds act directly on the envelope of bacterial cells, and potential mechanisms for the development of bacterial resistance to such compounds have already been identified [9,10]. The combination of various antibacterial compounds in one molecule is considered a promising approach for obtaining drugs that do not cause the development of resistance in bacteria [2,3,4].

We synthesized a hybrid of ciprofloxacin and cationic amphiphile DL_4_12, which can be referred to as DL_5_Cip6. The hybrid compound DL_5_Cip6 (Figure 1) contains in its structure two DABCO (D) residues connected by a pentamethylene linker (L_5_) and two ciprofloxacin residues connected to DABCO through a hexanoic acid-based linker (6). The synthesis and data on the antibacterial activity of DL_5_Cip6 were published earlier, and this compound showed high activity against *S. aureus.* The authors suggested that this effect is due to the high rate of DL_5_Cip6 penetration into cells [11]. Our study should clarify the exact mechanisms of the interaction of hybrid compound DL_5_Cip6 with cells of *S. aureus*.

When studying novel hybrid antibacterial compounds, first of all, their ability to kill bacteria is studied; then, the mechanisms of hybrids action are examined using various biochemical, microbiological, molecular biological, genetic and computer-aided drug design modeling methods [12,13,14,15]. Some publications contain data on transmission electron microscopy (TEM) of bacterial cells incubated with certain compounds for at least 14–16 h, more often a day. The illustrations, as a rule, show several cells: intact, damaged and completely destroyed [16,17,18,19]. We believe that such a long incubation does not allow us to detect “primary” changes in bacteria ultrastructure, which indicate the damage mechanisms of antibacterial compounds. Our experience has shown that the TEM of ultrathin sections makes it possible to understand mechanisms of the effects of antibacterial compounds on the example of cationic peptides and chlorhexidine against *S. aureus* [20] and *Candida albicans* [21].

With this in mind, we set the task of comparing the effect of the hybrid DL_5_Cip6 and its initial components, ciprofloxacin and DL_4_12, on the ultrastructure of *S. aureus* cells in order to identify the effects specific to each compound at the ultrastructural level. The ultrastructural features of ciprofloxacin and DL_4_12 action on *S. aureus* cells were determined in experiment, and the presence of these features under the influence of the DL_5_Cip6 hybrid was also revealed.

It is well known that existence in biofilm form makes *S. aureus* highly resistant to various agents, including antibacterial drugs [22]. To better assess the potential of our synthesized compounds to kill *S. aureus*, we investigated the effect of ciprofloxacin, DL_4_12, and DL_5_Cip6 on the *S. aureus* biofilm.

## 2. Materials and Methods

### 2.1. Antibacterial Compounds

Antibiotic ciprofloxacin hydrochloride (Zhejiang Guobang Pharmaceutical, Hangzhou, China) and two previously synthesized antibacterial compounds, DL_4_12 and hybrid DL_5_Cip6 (Figure 1) were used. Synthesis and antimicrobial properties of DL_4_12 and DL_5_Cip6 were published earlier [6,11].

Stock solutions of all compounds in DMSO (Serva, Heidelberg, Germany) had a concentration of 20 mmol/L (mM). Before being introduced into the bacterial culture, the compounds were diluted with Muller–Hinton broth (MHB) (OXOID, Basingstoke, UK).

### 2.2. Microorganism and Growth Conditions

The strain of *S. aureus* ATCC 25923 was stored at the Collection of Extremophile Microorganisms and Type Cultures of ICBFM SB RAS (Novosibirsk, Russia) at ‒70 °C in a NaCl-free Luria-Bertani (LB) (Difco, Franklin Lakes, NJ, USA) medium with 25% glycerol. Before experiments, the culture was plated on LB agar medium and incubated at 37 °C for 24 h.

### 2.3. Time-Dependence of Antibacterial Effect

The study was performed as described earlier [23]. An overnight culture of *S. aureus* was converted to the logarithmic growth phase by mixing with fresh MHB in a ratio of 1:50, cultivated for about 1 h until a cell concentration of 1 × 10^7^ CFU/mL was reached (OD_595_ = 0.1). The resulting cell suspension (50 µL) was mixed with ciprofloxacin, or DL_4_12, or DL_5_Cip6 at a concentration of 0.1, 0.01, 0.001, 0.0001 mM in a ratio of 1:1 and incubated at 37 °C in 96-well plates for 0, 30, 60, 120, 240, 300 min and 24 h, OD595 was measured. The experiment was carried out in triplicate.

### 2.4. Processing of S. aureus Suspension for TEM Examination

All reagents for TEM processing were purchased from EMS (Hatfield, PA, USA).

*S. aureus* culture in the logarithmic growth phase (concentration 1 × 10^7^ CFU/mL) was mixed with ciprofloxacin or DL_4_12, or DL_5_Cip6 (1:1) at a concentration of 0.1 or 0.01 mM. Then, the samples were incubated at 37 °C and 180 rpm (ES-20, BIOSAN, Riga, Latvia) for 15 and 45 min. In order to obtain cell mass necessary for TEM examination, the samples were prepared in 50 mL test tubes.

At the incubation end, 1 mL of 8% paraformaldehyde was added to 50 mL of the *S. aureus* suspension to stop the reaction. Then, the samples were centrifuged for 5 min at 6000 rpm (centrifuge SIGMA 2-16PK, Sartorius AG, Göttingen, Germany). The pellets were resuspended in 1 mL PBS and transferred into 2 mL tubes and centrifuged for 10 min at 4200× *g* using Eppendorf 5810R centrifuge (Eppendorf, Vienna, Austria). The supernatants were removed, and pellets were fixed in 2 mL of 4% paraformaldehyde in 0.2 M DMEM (Dulbecco′s Modified Eagle′s Medium) pH 7.2–7.4 for 24 h at 4 °C. Then, the samples were postfixed with 1% osmium tetraoxide solution for 1 h, routinely dehydrated in ethanol and acetone, and embedded in an epon–araldite mixture. The size of the pellets obtained made it possible to prepare one hard block from each sample.

To control the activity of ciprofloxacin, DL_4_12 and DL_5_Cip6 on *S. aureus* in a 50 mL-test tubes, 100 µL of cell suspension was taken from each tube, just before adding of paraformaldehyde (15 and 45 min incubation with compounds) as described in [23]. At the beginning of the time count, immediately after the addition of compounds, a “0” sample was taken. Since it took 5 min to prepare samples for titration, sample “0” was renamed to “5”. The number of viable cells in the suspension during incubation was determined by the number of colonies that grew after inoculation of 100 µL of samples on LB agar medium and 16–18 h incubation at 37 °C. Determining the number of grown colonies were carried out in triplicate.

### 2.5. Processing of S. aureus Biofilm Samples for TEM Examination

An overnight culture of *S. aureus* in trypticase soy broth (TSB) (Merck, Darmstadt, Germany) with 1% glucose was mixed with fresh TSB at a ratio of 1:50 and cultured until a cell concentration of 1 × 10^7^ CFU/mL was reached (OD_595_ = 0.1). Then, 2 mL of the resulting cell suspension were added to each well of 12-well plates with 3 mm thick layer of agar (Merck, Darmstadt, Germany), and incubated for 48 h at 37 °C (TSvL-80, Kasimov instrument plant, Kasimov, Russia) to form biofilms, as advised in [24]. The ciprofloxacin, or DL_4_12, or DL_5_Cip6 were added to the wells in 1 mL of TSB, the final concentration of the compounds was 0.1 mM and 0.5 mM.

Biofilms were incubated with compounds for 45 min, and then were fixed in plates by the Luft method, with a mixture of 1.2% solution of glutaraldehyde and 0.1% solution of ruthenium red in a ratio of 1:1 for 1 h [25,26]. After washing three times with DMEM, biofilms were additionally fixed in a mixture of 1% osmium tetroxide solution and 0.1% ruthenium red solution in a ratio of 1:1 for 2 h. After washing three times with DMEM medium, the biofilms together with the agar substrate were removed from the wells, and four rectangular pieces about 2 mm × 3 mm in size were cut from the central part of each film. Pieces of biofilm attached to agar were placed in 2 mL Eppendorf tubes, dehydrated according to the standard scheme, and embedded in an epon–araldite mixture. Four hard blocks were obtained from each biofilm.

To select the sites for targeted sharpening of the pyramids, semithin sections were prepared from the blocks of *S. aureus* biofilms using a Leica ultramicrotome (Leica, Wetzlar, Germany), and mounted on a glass slides. Semithin sections were obtained from three to four blocks of each biofilm, on each slide there were three consecutive sections. Semithin sections were stained with 1% Azur II solution (Lenreactive, Saint Petersburg, Russia) in 1% aqueous sodium pyroborate.

Semithin sections were examined in a Leica DM 2500 light microscope (Leica, Wetzlar, Germany), supplied with a Leica MC 170 HD digital camera (Leica, Wetzlar, Germany). Sites for sharpening pyramids were chosen on the semithin sections so that both cells and matrix were present on ultrathin sections (pyramids were cut out in the area where the matrix adjoined the cells).

### 2.6. TEM Analysis of S. aureus Cells and Biofilm

Ultrathin sections from all obtained blocks were prepared on an ultramicrotome EM UC7 (Leica, Wetzlar, Germany) using a diamond knife (Diatome, Nidau, Switzerland), and contrasted with 2% water solutions of uranyl acetate and lead citrate; some sections were left unstained. The sections were examined in a JEM 1400 TEM (JEOL, Tokyo, Japan). Digital images were collected by a Veleta side-mounted camera (EM SIS, Muenster, Germany). All measurements were made using the iTEM software version 5.2 (EM SIS, Muenster, Germany).

From each block, 20–25 ultrathin sections were obtained, which were examined in TEM. Each ultrathin section contained at least 1000 differently cut cells. Ultrathin sections have a thickness of 70–80 nm and can pass through cells in different planes, leading to an “imaginary” polymorphism of *S. aureus* cells. We analyzed cells and their structures on cross sections only. The thickness of the cell wall and cell membrane was measured only on sections perpendicular to these structures.

### 2.7. Processing of S. aureus Biofilm Samples for Light Microcopy

Biofilms of *S. aureus* grown on agar in 12-well plates, and treated with ciprofloxacin or DL_4_12, or DL_5_Cip6, for TEM studies, were fixed by the Luft method [25]. After washing three times with DMEM medium, all the liquid was removed from the wells, and 100 µL of warmed liquid TSB agar were added to each well. After the agar solidified, 2 mL of 4% paraformaldehyde was added to the wells and left for a day at 4 °C. Thus, the biofilms were covered with agar layers on both sides, like a sandwich, to avoid loss of the cells and matrix.

The “sandwich” biofilms were removed from the wells, and were dehydrated using a Tissue-Tek II histological machine (Sakura Finetek Europe, Alphen aan den Rijn, Netherlands). Biofilms of *S. aureus* together with agar layers were cut in half, oriented downwards, and embedded into blocks with Histomix medium (Biovitrum, Saint Petersburg, Russia). Paraffin sections about 3.5 μm of thickness were made on a Leica RM 2255 microtome (Leica, Wetzlar, Germany) from blocks, mounted on glass slides and dried in air.

Sections were deparaffinized and stained for 10 sec with Azur II (Lenreactive, Saint Petersburg, Russia), and embedded in BioMount medium (Biovitrum, Saint Petersburg, Russia). The sections were examined under a Leica DM 2500 light microscope (Leica, Wetzlar, Germany); images were taken with a Leica MC 170 HD digital camera (Leica, Wetzlar, Germany). The measurements of biofilm parameters were made using the AxioVision 4.8 software package (Zeiss, Oberkochen, Germany). Three sections were examined and at least 20 measurements were made from each biofilm.

## 3. Results

### 3.1. Features of S. aureus Sample Preparation for TEM

The concentrations of the ciprofloxacin, DL_4_12 and DL_5_Cip6 used for the TEM samples preparation were selected based on the results of studying the kinetics of their effects (Appendix A), and amounted to 0.01 mM and 0.1 mM.

To reveal the initial effects of ciprofloxacin, DL_4_12 and DL_5_Cip6 on the ultrastructure of *S. aureus* cells, the cell suspension was incubated with the compounds for 15 and 45 min. The time of sampling for TEM studies was chosen based on our studies of the effect of antibacterial peptides on cells of the same strain of *S. aureus* [20].

To assess the “rate” of *S. aureus* killing by ciprofloxacin, DL_4_12 and DL_5_Cip6, we studied the changes in viable cells concentration during incubation with compounds for selected time intervals (Table 1). The data presented in Table 1 indicate the different “killing rate” of the compounds used. DL_4_12 proved to be a fast and effective killer, while ciprofloxacin and DL_5_Cip6 acted similarly, the decrease in cells number after 45 min of incubation was insignificant. These data confirmed the correctness of sampling after 15 and 45 min of interaction of the bacterium with the compounds, since *S. aureus* cells at different stages of damage are required for ultrastructure analysis.

### 3.2. Ultrastructure of Intact Planktonic S. aureus Cells

Intact *S. aureus* cells on ultrathin sections had rounded shape and 600–700 nm diameter (Figure 2A,B). Cells were actively dividing, 35–40% of cells showed a division septum at different stages of formation (Figure 2A,C and Appendix A).

The cell envelope of *S. aureus* was composed of a cell wall and a cell membrane separated by an intermediate layer (Figure 2A,C, Figure 3A and Appendix A). The cell wall had the same average electron density along its entire length and a thickness of 17–18 nm; the outer surface could be smooth or slightly rough. The width of the intermediate layer is 2–3 nm; the layer contained a granular substance of high electron density (Figure 2C and Figure 3A). The intermediate layer in some publications is called “periplasm” [27]. The term “intermediate layer” will be used in this work.

The cell membrane was tightly bound with the cytoplasm and the granular substance of the intermediate layer; the electron-dense layers of the membrane fused with them and became indistinguishable on ultrathin sections. A membrane middle layer of low electron density (3–4 nm thickness) was clearly visible on ultrathin sections (Figure 2C). We did not observe mesosomes, which were often present in *S. aureus* cells of used strain when cultivated in saline [20].

The irregularly shaped nucleoid zone usually was located in cell center, and differed from the cytoplasm by a lower electron density and the absence of granularity (Figure 2A,B). Strands of DNA were difficult to recognize, and therefore the nucleoid was poorly visible on TEM images; this phenomenon was mentioned in many studies [27,28,29,30]. The cytoplasm (Figure 2A–C) of medium electron density contained ribosomes and looked the same in most cells. The cytoplasm adjacent to the cell membrane was devoid of ribosomes.

### 3.3. The Effect of Ciprofloxacin on S. aureus Ultrastructure

Ciprofloxacin realizes its antimicrobial action by inhibition of DNA gyrase, essential for chromosome replication and function. Disrupting the structure of DNA gyrase enzyme leads to the suppression of bacterial division [2,31]. We observed clear signs of impaired cell division in *S. aureus*, incubated with ciprofloxacin for 45 min: the number of cells with a septum was significantly less than in intact culture (only 10–15% of cells vs. 35–40%), and visible nucleoids were rare (Appendix A). Distinct changes in the ultrastructure of *S. aureus*, caused by exposure to ciprofloxacin, were observed only after 45 min of incubation: in all cells, disturbances of varying degrees were visualized, although completely destroyed cells were rare (Appendix A).

Obviously, in order to penetrate into *S. aureus* cytoplasm and affect its target, ciprofloxacin must overcome the cell envelope. The thickness of the cell wall and its electron density did not visually change throughout the entire period of incubation (Figure 2D–F), which indicates the passage of ciprofloxacin through the wall substance without disturbing it at TEM level. “Bubbles” and “rods” about 10 nm in size from a medium electron density material appeared on some areas of cell wall surface, giving it a “shaggy” appearance (Figure 2D–F). The presence of these structures obviously reflects the release of some substances by *S. aureus* cells under the influence of ciprofloxacin.

The cell membrane, tightly connected to the intermediate layer, became wavy and folded, small white folds are clearly seen in perpendicular sections of *S. aureus* cells (Figure 2E). Very small electron-dense “grains” (about 1–2 nm) were observed in some areas of the intermediate layer (Figure 2D), reflecting the violation of its integrity. Large folds of the cell membrane and the accompanying intermediate layer were found in many cells, giving the cytoplasm a scalloped shape on sections and reflecting cell deformation under the action of ciprofloxacin (Figure 2D). The degree of ultrastructural changes differed between cells, as well as between areas of the same cell.

Ciprofloxacin treatment changed the appearance of the cytoplasm, which lost its granularity and looked motley and cloggy (Figure 2D–F). These changes were mainly due to low electron density spheres formed at the border of the cytoplasm and cell membrane (Figure 2F). The spheres migrated to the electron-dense cytoplasm, where they lost their distinct shape and became blurry.

### 3.4. The Effect of DL_4_12 on S. aureus Ultrastructure

The DL_4_12 showed a pronounced damaging effect on a suspension of *S. aureus*: a large number of destroyed cells and cell debris were seen in ultrathin sections (Appendix A). This corresponds to the fast destruction of *S. aureus* cells revealed by microbiological methods (Appendix A, Table 1). The ultrastructure of *S. aureus* cells incubated for 15 and 45 min with the DL_4_12 did not noticeably differ, the nature of the changes in the cells was equal; an increase in the concentration of DL_4_12 from 0.01 to 0.1 mM increased the degree of changes. Figure 3 shows the development of the most striking changes in the structure of *S. aureus* cells under the influence of DL_4_12.

A distinctive feature of the DL_4_12 effect was fast (less than 15 min) cell wall thickening up to 40 nm or more compared to 17–18 nm in intact cells (Figure 3A,B and Appendix A). Cell wall thickening was focal at concentration of DL_4_12 0.01 mM, and involved the entire wall surface when the concentration increased to 0.1 mM. At the next stage, the surface of the cell wall became “bumpy” or “shaggy”, it lost its clear outline, swells and become uniformly electron-transparent (Figure 3C,D). These changes unequivocally indicate that the DL_4_12 directly interacted with cell wall and caused severe damage to its structure.

Another striking sign of DL_4_12 effect was the appearance of a material of high electron density in *S. aureus* cells, obviously reflecting changes in the components of the cytoplasm when DL_4_12 penetrated the cells. The electron-dense structures in unstained ultrathin sections showed contrast due to the binding to osmium tetroxide and the properties of the high-contrast TEM digital camera. The absence of contrasting heavy residues of uranyl and lead greatly reduced the visibility of cellular structures, but allowed us to see fine details of electron-dense material.

At the first stage, electron-dense structures appeared on the periphery of *S. aureus* cells and have a small sizes (Figure 3E); as it accumulated, their size increased, and they took the form of a more or less electron-dense “pancake” with jagged edges adjacent to the cell envelope (Figure 3E–H). Obviously, as the material altered the cell envelope structure, its components became indistinguishable. Structural disturbances do not allow visualization of damage to the cell membrane and intermediate layer (Figure 3C,D,J and Appendix A). Electron-dense material spread deep into the cell, acquiring a variety of shapes and increasing in size, up to bizarrely shaped bodies (up to 200 nm), connected to the cell membrane, or located near it. In some sections, fine filaments were seen in electron-dense material (Figure 3I–K and Appendix A).

Pronounced consequences of the entry of DL_4_12 into the cytoplasm of *S. aureus* were observed after 15 min of incubation. The intermediate layer expanded to 5–6 nm (2–3 nm in intact cells); its material loosened and showed grain structure (Figure 3A,B and Appendix A). Noticeable changes in the cell membrane were detected when an electron-dense substance appeared under it. It is very likely that the membrane is rapidly damaged at the molecular level and these changes cannot be visualized in TEM.

*S. aureus* cells continued actively divide against the background of DL_4_12, despite pronounced cytoplasm alteration; the ratio of cells with and without division septum (Appendix A) did not change (35–45% cells have division septum), regardless of the compound concentration and time of incubation.

The results obtained evidence for DL_4_12 quick and strong effect of *S. aureus* cells, leading to their destruction. The first visible target for DL_4_12 was the cell wall; however, we did not find violations of its integrity despite of a pronounced thickening: there were no “cracks” and “channels” larger than 2 nm (the “working” resolution of TEM). We suppose that the impact of DL_4_12 on *S. aureus* cell wall was the main cause for the observed rapid destruction of a significant proportion of bacteria cells.

### 3.5. The Effect of DL_5_Cip6 on S. aureus Ultrastructure

We expected to find in *S. aureus* cells incubated with DL_5_Cip6 all the changes characteristic of its constituents, ciprofloxacin and DL_4_12. However, when bacteria were incubated with DL_5_Cip6 for 15 and 45 min, signs of division suppression and rapid cell destruction caused by ciprofloxacin and DL_4_12 were not detected, regardless of the dose used (Appendix A).

Figure 4 shows representative illustrations of ultrastructural changes in *S. aureus* cells incubated with DL_5_Cip6. Figure 4A–E demonstrates changes that can be attributed to the ciprofloxacin effect, and Figure 4F–H those that can be attributed to the DL_4_12 effect.

The impact of DL_5_Cip6 on *S. aureus* led to loosening of the cell wall and loss of its outer border clarity. Changes in the wall developed gradually, the images show areas of a smooth wall and areas of pronounced alteration (Figure 4). At the same time, cell wall thickening, which was a feature of DL_4_12 effect, was absent.

The cell membrane of *S. aureus* was affected by DL_5_Cip6, and these changes were similar to those observed in bacteria treated with ciprofloxacin, but were of a higher degree. The cell membrane, visualized by the electron-transparent layer, lost its structure and, together with the adjacent intermediate layer, formed small folds (Figure 4A), like those in ciprofloxacin treated cells. Then, folds of the cell membrane increased in size and formed structures resembling a “bowl”, which contained an electron-dense material surrounded by a membrane (Figure 4B). We observed different varieties of “bowl”-like structures, containing structureless material, probably destroyed cytoplasm. All “bowl”-like structures were clearly separated from cytoplasm (Figure 4C–E). In the image of a cell with minor structural disturbances, strands of a substance of medium electron density are visible in the cell wall covering the “bowl” (Figure 4C). Figure 4D,E show cells with clumpy cytoplasm and altered cell wall, fragments of membranes and electron-dense material are visible in the bowl area.

The loss of structure by the *S. aureus* cytoplasm upon ciprofloxacin treatment began with the formation of low electron density spheres at the interface between the cytoplasm and the cell membrane (Figure 2D–F). The same structures were observed in *S. aureus* cells exposed to DL_5_Cip6 (Figure 4A; several low electron density spheres, and Figure 4D,E; blurred spheres of low electron density).

The clearest manifestation of the effect of DL_4_12 in the composition of the DL_5_Cip6 molecule was the appearance and accumulation of an electron-dense material along the periphery of the cytoplasm (Figure 4F–H). Electron-dense material in *S. aureus* cells, when exposed to DL_5_Cip6, first settled down as small clumps that formed a discontinuous layer near the cell membrane (Figure 4F,G). Then, the material accumulated and, as a rule, spread into the cytoplasm, giving it a high electron density (Figure 4H). However, clearly defined electron-dense bodies, as under the action of DL_4_12, were not formed. The cytoplasm looked “stained”; an electron-dense substance, as such, was not visible, in contrast to the effect of DL_4_12, which indicates a different interaction of the DL_5_Cip6 and DL_4_12 molecules with the cytoplasm.

Our study revealed that the action of DL_5_Cip6 on *S. aureus* combined the effects of its components, ciprofloxacin and DL_4_12, but this combination is not a simple sum. The effect of ciprofloxacin dominated in ultrastructure changes in cells treated with DL_5_Cip6, although the hybrid did not suppress cell division as ciprofloxacin. In general, the degree of injury under the influence of DL_5_Cip6 was higher than in ciprofloxacin.

### 3.6. The Effect of Ciprofloxacin, DL_4_12 and DL_5_Cip6 on S. aureus Biofilm Morphology

We have identified destructive changes in planktonic cells of *S. aureus* when exposed to ciprofloxacin, DL_4_12 and DL_5_Cip6. To assess the ability of these compounds to damage *S. aureus* biofilm, we decided to apply the method of light microscopy of paraffin sections. The paraffin sections were obtained passing through the center of *S. aureus* biofilms, which made it possible to evaluate the effect of DL_4_12, ciprofloxacin, and DL_5_Cip6 on the entire biofilm, and not on its small areas.

An intact *S. aureus* biofilm consists of a layer of cells (stained bright blue with Azur-II) and an adjacent matrix (stained yellow-brown with ruthenium red) (Figure 5A). The thickness of the paraffin section was about 3 μm; therefore, in the section, *S. aureus* cells (600–700 nm in diameter) can lie in several rows, forming visible continuous mass, the boundaries between cells were poorly distinguishable. The thickness of the cell layer in the *S. aureus* biofilm was approximately the same along the entire length of the biofilm (20–25 µm); in some places, the layer thickened to 40 µm or thinned to 15 µm; empty cavities 2–5 μm in size were also observed in the cell layer.

The structure of the *S. aureus* biofilm observed on paraffin sections corresponded to its morphology described in other works [32]. The matrix layer in *S. aureus* was located on the biofilm outer surface, bordering the environment, and generally tightly adjacent to the cells. The thickness of the matrix layer varied in different parts of the biofilm and reached 30 μm. The matrix consisted of small (300–400 nm) rounded globules (Figure 5A).

The cell layer images in Figure 5A, showing the morphology of *S. aureus* intact biofilms at different magnifications, appear darker than subsequent images of biofilms incubated with DL_4_12 and DL_5_Cip6. These differences are not associated with staining errors, but are due to a decrease in the number of stained cells in the thickness of the paraffin section. We strictly ensured that the thickness of the sections adjusted by the microtome was the same, so we can state that the intensity of the blue color reflects the concentration of cells within a three-micron section.

Ciprofloxacin caused pronounced changes in both the cell layer and matrix of *S. aureus* biofilm compared to the intact biofilm; the degree of change increased with increasing concentration (Figure 5B,C). The layer of cells was uneven and loose, its thickness varied from 5 to 25–30 µm and thinned in comparison with intact biofilm (thickness is up to 40 µm). These changes, together with a decrease in biofilm staining intensity, indicate a decrease in the number of *S. aureus* cells in biofilm. Ciprofloxacin significantly reduced the mass of the matrix; instead of a continuous layer, only small accumulations (up to 15 µm in diameter) were observed (Figure 5B,C).

Incubation of the *S. aureus* biofilm with DL_4_12 led to loosening of the cell layer and an increase in the size of cavities between cells (up to 10–15 μm) (Figure 5D,E), which made it impossible to accurately determine its thickness. The matrix was loosened and separated from the cell layer in many places (Figure 5). The degree of visual changes in the biofilm under the action of DL_4_12 did not depend on its concentration.

The cell layer of the *S. aureus* biofilm treated with DL_5_Cip6 became loose and had large cavities; the matrix layer disintegrated and was represented by separate fragments (Figure 5F,G). In general, the pattern of damage to *S. aureus* biofilm by DL_5_Cip6 was similar to that of DL_4_12, but the hybrid showed a concentration dependence.

### 3.7. The Effect of Ciprofloxacin, DL_4_12 and DL_5_Cip6 on S. aureus Biofilm Matrix

Pronounced changes in the *S. aureus* biofilm matrix, clearly visible in a light microscope, required study of matrix in TEM. We applied ruthenium red for visualization of a matrix, because standard TEM-processing is unable to preserve the matrix, and examined the same biofilms treated with ciprofloxacin, DL_4_12 and DL_5_Cip6.

The matrix of intact *S. aureus* biofilm (Figure 6A–C) was composed of globules (150–300 nm) with an electron-dense center (50–100 nm) surrounded by a layer of medium electron density (50–100 nm). The periphery and center were clearly distinguishable; the ratio between them varied in different globules. The globules were usually located in the form of clusters of various shapes, sometimes in the form of branched chains of 10–15 globules. Individual globules were rare and had a spherical shape; few globules contained several centers. The outer surface of the globule on perpendicular sections formed a clear border, which had a membrane-like appearance in some areas (Figure 6A).

The shape of the central part varied depending on the location of the globules and the plane of the section (Figure 6A–C). Ribbon-like structures of high electron density stand out in the granular substance of the central part of the globule and contact with each other, forming intricate images (Figure 6C). The peripheral layer consisted of filaments (2 nm diameter) of medium or low electron density, which could be arranged radially (Figure 6B).

Thus, the globules that form *S. aureus* biofilm matrix have a complex ultrastructure that reflects their rich molecular composition. Ciprofloxacin and DL_4_12 and DL_5_Cip6 interacted with biofilm incubated for 48 h, and we believe that the observed changes in the ultrastructure of the matrix reflect the effect of these compounds on the “mature” matrix.

Incubation with ciprofloxacin led to the deformation of the globule and the loss of clarity of the outer surface, which became “shaggy” (Figure 6D). In some globules, the thickness of the outer layer decreased to 3–5 nm (Figure 6E). Most of the globules were located singly, rarely uniting in short chains (3–4 globules each).

The action of DL_4_12 manifested itself by a decrease in the size of most globules to 100–150 nm and of the peripheral layer thickness to 10 nm, while the structure of the central part remained unchanged (Figure 6F). The rest of the globules retained the dimensions inherent in the intact matrix; although the area of the central part significantly decreased, it was deformed and acquired an irregular shape. The central part was represented by electron-dense grains arranged in chains and surrounded by a few filaments of medium electron density (Figure 6G).

The action of the DL_5_Cip6 hybrid led to loosening of the peripheral layer of globules and a decrease in its thickness; many globules fused together (Figure 6H,I). The central part of the globules became heterogeneous and lumpy. The electron-dense material in its composition was represented by rare single grains (Figure 6I).

The biofilm matrix *S. aureus* does not surround each cell [33], but lies in a layer on the biofilm surface. The molecular composition of the *S. aureus* biofilm matrix has been studied in sufficient detail [34,35,36]; however, information on the ultrastructure of the matrix has not been published.

Our TEM study, for the first time, showed that *S. aureus* biofilm matrix consists of globules with a complex structure. All antibacterial compounds used in this work caused pronounced and different changes in the structure of globules, which indicates a direct interaction of molecules with matrix molecules.

## 4. Discussion

The problem of fighting bacterial infections that take millions of lives is becoming more acute every year. Antibiotics are rapidly losing their key positions in the infections treatment under the pressure of new resistant bacteria strains and against the background of their improper use [1,2,3]. In a huge array of developments of new antibacterial drugs, antimicrobial peptides (AMP), which have the widest spectrum of biological activity, occupy leading positions [37,38]. With all the undeniable promise of AMPs, there are objective limitations to their use: for example, in the body, regardless of the route of administration, AMPs are exposed to proteolytic enzymes [37]. This circumstance, among others, determines the need to create synthetic compounds having similar antibacterial properties, but devoid of AMP’s shortcomings. Detailed analyses of various classes of synthetic compounds that can be used to solve this problem are given in [39].

Previously, we synthesized a compound “DL_4_12” (Figure 1), in which quaternary residues of 1,4‒diazabicyclo[2.2.2]octane (DABCO) were connected by tetramethylene linker with a dodecane substituent, and showed its high antibacterial activity [6]. This class of compounds was not mentioned in the detailed review by Stojkovic et al. [39], published in May 2023. In this work, for the first time, we showed that the polycationic amphiphile DL_4_12 directly interacts with the *S. aureus* cell wall, starting a pronounced disruption of its structure. Direct interaction of synthetic polycationic AMPs with bacteria cell wall was predicted by computer simulations [40], but not shown in a real experimental system. We visualized such contact and thus experimentally confirmed the existence of this mechanism.

The hybrid antibacterial compounds are considered as promising agents to overcome the antibiotic resistance of bacteria [2]. However, this conclusion emerges from theoretical calculations based on a simple summation of the known effects of hybrid components. Meanwhile, our previous study of hybrids of ciprofloxacin and tetracationic derivatives of DABCO showed the important role of the linker in determining the biological properties of the hybrid [11]. An analysis of publications shows that the mechanisms of hybrid molecules effect on bacterial cells have not been experimentally studied.

In this work, for the first time, we examined interaction of a hybrid DL_5_Cip6 with *S. aureus* cells by TEM of ultrathin sections. For correct analysis of the hybrid effects, we first studied in detail the ultrastructural characteristics of the interaction of its components, ciprofloxacin and DL_4_12, with *S. aureus* cells. The study of the interaction of the hybrid DL_5_Cip6 with *S. aureus* cells, to our surprise, revealed the predominance of ciprofloxacin effects in the ultrastructural picture; manifestations of the polycationic amphiphile DL_4_12 were less pronounced. Analysis of the *S. aureus* cell ultrastructure under the influence of the hybrid DL_5_Cip6 made it possible to determine specific signs of cell structure disturbances, which are not a simple sum of the effects of its components, DL_4_12 and ciprofloxacin. Obviously, the nature of the linker between the initial components in the hybrid molecule plays an important role, which requires additional study.

Unfortunately, we cannot compare our results on studies of hybrid DL_5_Cip6 and its initial compounds with similar ones and discuss them correctly, since the relevant papers have not been published. To our surprise, no ultrastructural changes in bacteria have been described even under the influence of the long-known ciprofloxacin.

Although our main task was not to compare the antibacterial potential of the studied compounds against *S. aureus* cells, their high destructive activity against planktonic cells prompted us to investigate their effect *S. aureus* biofilm. Using a simple method for assessing the state of the biofilm on paraffin sections, we found that the damaging effect of the compounds developed quite quickly. The data obtained evidence that light microscopy can be applied for evaluation of antibacterial effects on *S. aureus* biofilm.

The molecular composition of the *S. aureus* biofilm matrix has been studied in sufficient detail [33,35,36]; however, information on the ultrastructure of the matrix has not been published. The biofilm matrix *S. aureus* does not surround each cell [33], but lies in a layer on the surface of the biofilm cells. Our TEM study showed that *S. aureus* biofilm matrix consists of globules with a complex structure, obviously reflecting the multicomponent composition of the matrix. All antibacterial compounds used in this work caused pronounced and different changes in the structure of globules.

## 5. Conclusions

In this work, we examined ultrastructural changes in *S. aureus* cells in response to the hybrid DL_5_Cip6 and its constituents, ciprofloxacin and polycationic amphiphile DL_4_12. Ultrastructural changes in *S. aureus* cells developed within 45 min of ciprofloxacin are described.

It has been shown, for the first time, that a polycationic amphiphile DL_4_12 directly interacts with the *S. aureus* cell wall, which serves as the primary target for this compound. The ultrastructure of *S. aureus* treated with hybrid is a combination of the changes that develop under the influence of ciprofloxacin and DL_4_12, with clear predominance of signs of ciprofloxacin.

*S. aureus* biofilm can be examined on paraffin sections, which allows assessment of damage to the cell layer and matrix by antibacterial compounds. For the first time, we showed, using TEM of ultrathin sections, that *S. aureus* biofilm matrix is composed of globules with a complex structure. Globule ultrastructure changes differently under treatment with ciprofloxacin, polycationic amphiphile and their hybrid DL_5_Cip6.

## Figures and Tables

**Figure 1 microorganisms-11-02192-f001:**
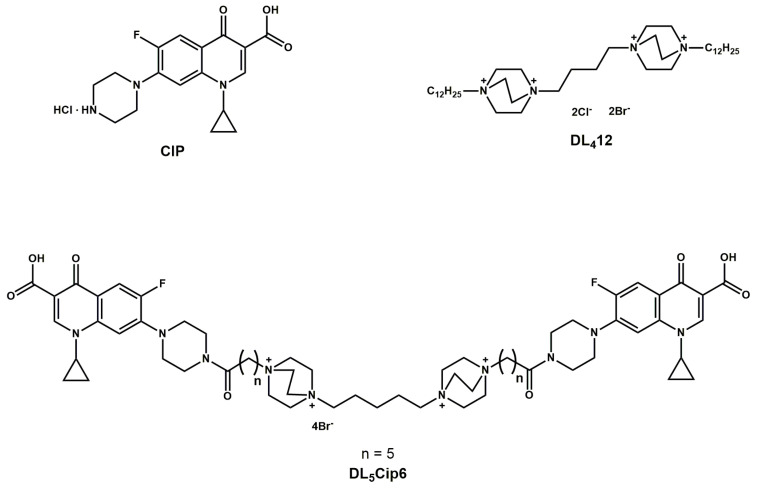
Chemical structures of ciprofloxacin (Cip); cationic amphiphile DL_4_12 (where D is DABCO (1,4‒diazabicyclo[2.2.2]octane)), L_4_ is a tetramethylene linker, 12 is a dodecyl residue); DL_5_Cip6 (where D is DABCO, L_5_ is a pentamethylene linker, Cip is ciprofloxacin, and 6 is a hexanoic acid linker).

**Figure 2 microorganisms-11-02192-f002:**
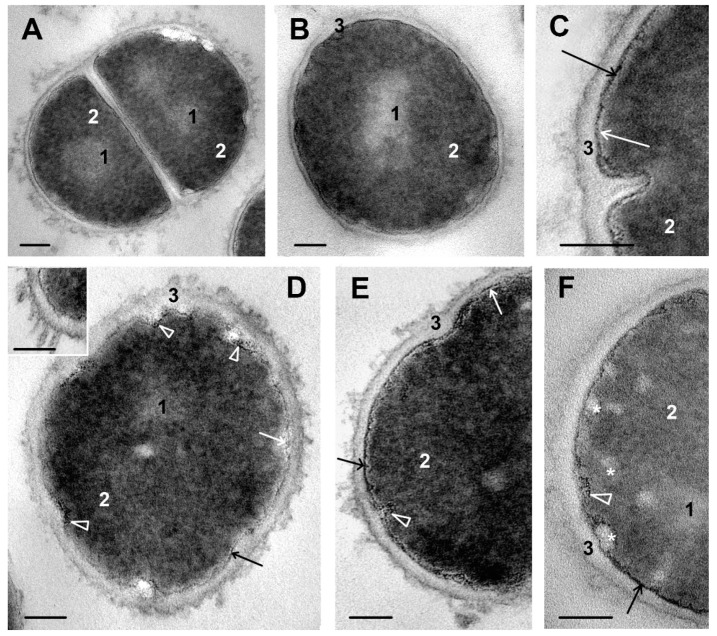
Changes in ultrastructure of *S. aureus* cells incubated with ciprofloxacin. (**A**–**C**)—Representative images of intact *S. aureus* cells. (**A**)—A cell with division septum; (**B**)—single cell; (**C**)—formation of division septum. (**D**–**F**)—*S. aureus* cells incubated with ciprofloxacin for 45 min. (**D**)—A cell demonstrating all detected kinds of damage by ciprofloxacin; insert shows release of a material on the cell wall surface. (**E**,**F**)—Damage of cell envelope and cytoplasm. 1—nucleoid; 2—cytoplasm; 3—cell wall; white arrows show cell membrane; black arrows—intermediate layer; arrowheads show sites of intermediate layer alteration; asterisks—spherical structures adjacent to cell membrane and in cytoplasm. The length of the scale bars corresponds to 200 nm (**A**,**C**,**E**); 100 nm (**B**,**D**,**F**). TEM, ultrathin sections.

**Figure 3 microorganisms-11-02192-f003:**
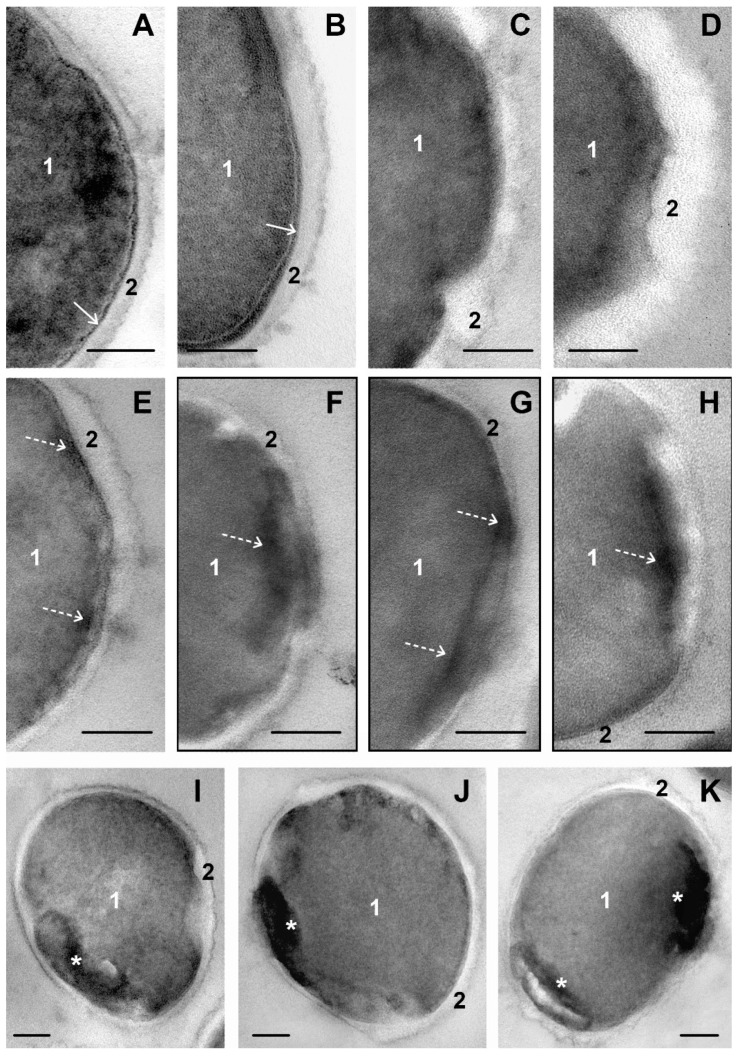
Changes in ultrastructure of *S. aureus* cells incubated with cationic amphiphile DL_4_12 for 45 min. (**A**–**D**)—Successive cell wall changes: (**A**)—intact cell; (**B**)—cell wall thickening; (**C**,**D**)—swelling and loss of cell wall structure. (**E**–**H**)—Expansion of electron-dense material into the cell, disruption of the cell envelope and cytoplasm; black frames border non-contrasted sections. (**I**–**K**)—The nature of accumulations of electron-dense material inside cells, damage to cytoplasm. 1—cytoplasm; 2—cell wall; white arrows point to intermediate layer; broken arrows—electron-dense material; asterisks—accumulations of electron-dense material. TEM, ultrathin sections. The length of the scale bars corresponds to 100 nm.

**Figure 4 microorganisms-11-02192-f004:**
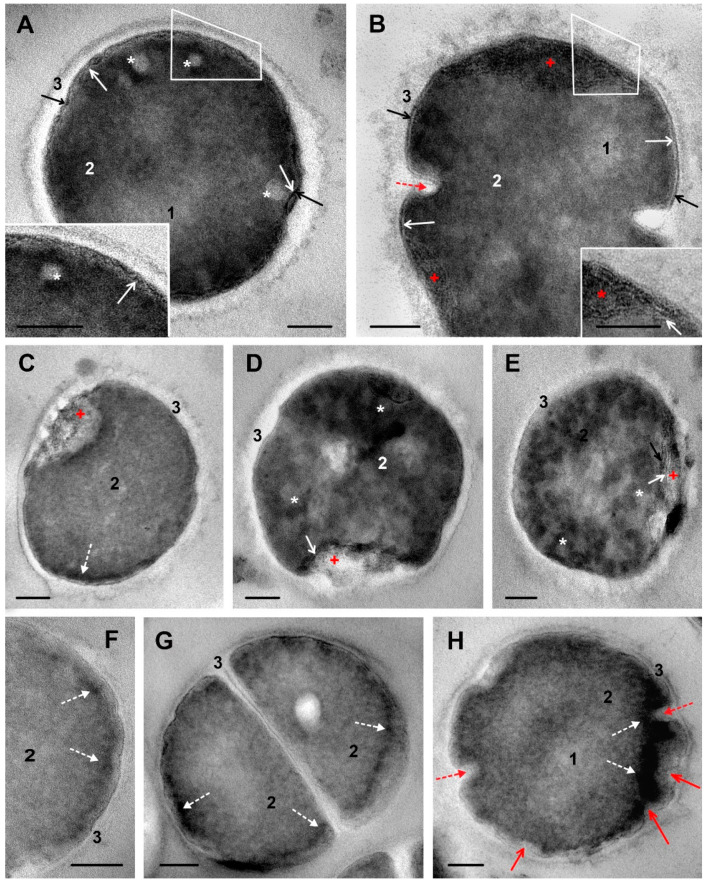
Changes in ultrastructure of *S. aureus* cells incubated with hybrid DL_5_Cip6. (**A**)—prominent folding of cell membrane and damage of intermediate layer; insert: enlarged area inside a contour. (**B**)—a “bowl”-like structures composed of cell membrane folds, which enclose electron-dense material; insert: enlarged area inside a contour. Note alteration of cell wall structure. (**C**–**E**)—variants of “bowl” shape. (**C**)—a cell with minor structural disturbances, cell wall covering a “bowl” contains strands of medium electron density. (**D**,**E**)—cells with clumpy cytoplasm and altered cell wall; “bowl” areas contain membrane fragments and electron-dense material. (**F**–**H**)—the appearance and accumulation of electron-dense material in *S. aureus* cells. (**F**)—small clumps; (**G**)—electron-dense material in the cytoplasm along cell periphery; (**H**)—“staining” of cytoplasm by electron-dense material. 1—blurred area of nucleoid; 2—cytoplasm; 3—cell wall; white arrows show cell membrane; black arrows—intermediate layer; asterisks—spherical structures in cytoplasm; red arrow—folds of cell membrane and intermediate layer; broken red arrow—septum formation; red cross—“bowl”-like structure; broken white arrows—electron-dense material; red arrows—deformation of cytoplasm. TEM, ultrathin sections. The length of the scale bars corresponds to 100 nm.

**Figure 5 microorganisms-11-02192-f005:**
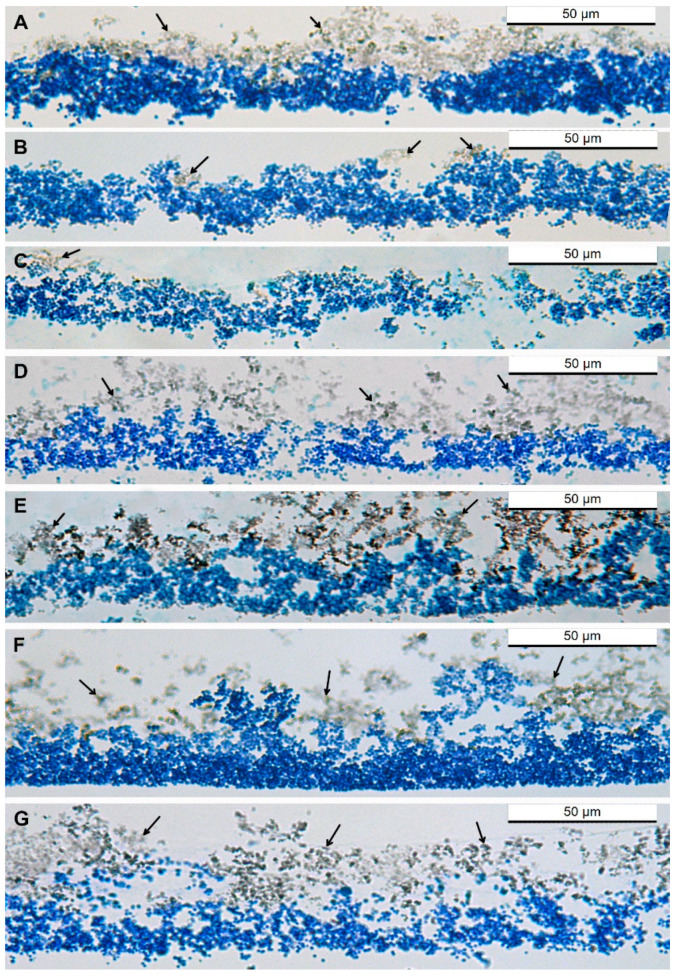
Changes in morphology of *S. aureus* biofilm incubated with antibacterial compounds for 45 min. (**A**)—intact biofilm. Incubation with ciprofloxacin: (**B**)—0.1 mM and (**C**)—0.5 mM. Incubation with DL_4_12: (**D**)—0.1 mM and (**E**)—0.5 mM. Incubation with a DL_5_Cip6: (**F**)—0.1 mM and (**G**)—0.5 mM. Cells of *S. aureus* are stained in blue. The arrows show the brown matrix. Luft fixation. Paraffin sections, Azur II staining, light microscopy.

**Figure 6 microorganisms-11-02192-f006:**
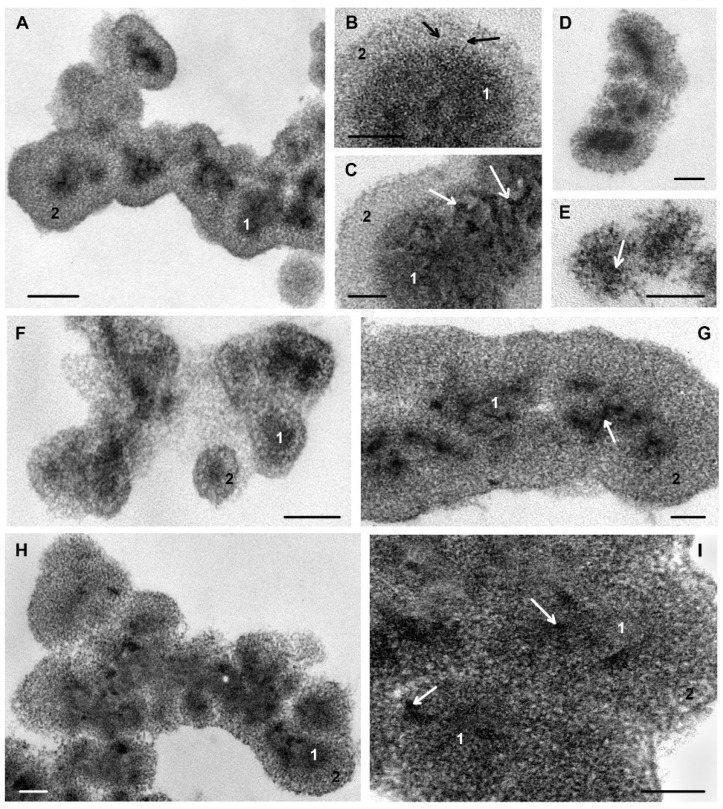
Effect of antibacterial compounds on ultrastructure of *S. aureus* biofilm matrix. (**A**–**C**)—globules of intact matrix; (**D**,**E**)—incubated with ciprofloxacin; (**F**,**G**)—incubated with DL_4_12; (**H**,**I**)—incubated with DL_5_Cip6. 1—central part of the globule, 2—peripheral part of the globule. White arrows show electron-dense grains; black arrows show filaments of the peripheral layer. Luft fixation. TEM, ultrathin sections. The length of the scale bars corresponds to 100 nm.

**Table 1 microorganisms-11-02192-t001:** Changes in concentration of *S. aureus* cells (CFU/mL) incubated with antibacterial preparations. The data are presented as lg (CFU/mL). The concentration of compounds is given in µM/mL.

*S. aureus* Incubation Time	Intact Cells	DL_4_12 0.01	DL_4_12 0.1	DL_5_Cip6 0.01	DL_5_Cip6 0.1	CIP 0.01	CIP 0.1
5 min	6.95 ± 0.07	3.07 ± 0.11	3.05 ± 0.11	6.92 ± 0.16	6.60 ± 0.05	6.99 ± 0.23	6.95 ± 0.05
15 min	7.14 ± 0.09	2.98 ± 0.03	2.75 ± 0.4	6.21 ± 0.18	4.41 ± 0.16	7.01 ± 0.13	6.68 ± 0.25
45 min	7.25 ± 0.10	2.85 ± 0.11	1.22 ± 0.19	6.39 ± 0.19	3.76 ± 0.19	6.45 ± 0.26	6.48 ± 0.18

CIP—ciprofloxacin. DL_4_12—D is DABCO (1,4‒diazabicyclo[2.2.2]octane)), L_4_ is a tetramethylene linker, 12 is a dodecyl residue. DL_5_Cip6—D is DABCO, L_5_ is a pentamethylene linker, Cip is ciprofloxacin, and 6 is a hexanoic acid linker.

## Data Availability

Research data are available in tables and figures from the manuscript and Appendix A.

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
