# Peer review of "Changes in the Ultrastructure of Staphylococcus aureus Cells Make It Possible to Identify and Analyze the Injuring Effects of Ciprofloxacin, Polycationic Amphiphile and Their Hybrid"

_microorganisms, 2023, doi:10.3390/microorganisms11092192_

Round 1

Reviewer 1 Report

The manuscript by Grigoreva et al examined the ultrathin sections of S. aureus cells and biofilm structure by transmission electron microscopy (TEM) to understand the antibacterial effects of hybrid compounds composed of ciprofloxacin and polycationic amphiphile based on 1,4-diazabicyclo [2.2.2] octane derivatives (DL412) (DABCO). The authors initially determined the MIC of all three compounds (Cip, DL412 and DL5Cip6) with two different bacterial concentrations (105 and 108 cfu/ml). Based on the MIC of these three compounds, the authors used 0.1 µM and 0.01 µM of antibiotics or hybrid compounds on bacterial culture and performed TEM on 15 min and 45 min bacterial culture. TEM analysis showed differential cell damage with these three antibacterial compounds. The authors also studied the effect of the hybrid and its compounds on the S. aureus biofilm morphology by TEM and biofilm matrix by light microscopy. 

Although the manuscript highlights the importance of TEM to study any specific antibiotic or antibacterial compounds on bacteria in the cellular level, the lack of different concentrations and incubation time limits the significance of the findings. The authors should use the MIC, 1/2MIC, 1/4MIC of a given antibiotic or antibacterial compound and take images of the cells at various time points (incubation time). The authors also should include TEM images for overnight incubation culture to show the end result of the damaged cells. The authors should include another bacterium, especially a Gram-negative bacterium to see the effect of this hybrid compound. Despite these major suggestions, I have a few specific comments on the manuscript for the authors.

1.     Please separate the results and the discussion section for a better presentation. Currently, the results are too descriptive. Use symbols and numbers to describe the figures. In the discussion section, compare these findings with the related literature of similar studies.

2.     Use the full organism name in the title of the manuscript. Also, italicize the scientific name throughout the manuscript where applicable.

3.     Instead of mentioning ‘Cip’, state the full name of the antibiotic (ciprofloxacin) throughout the manuscript.

4.     From the MIC values, it shows that the antibacterial activity is highest for Cip (0.00125 µM) followed by DL412 (0.01 µM) and DL5Cip6 (>2 µM) when the initial bacterial concentration is 105 cfu/ml. In the case of 108cfu/ml, the antibacterial activity is highest for DL412 (0.01 µM), followed by Cip (>1 µM) and DL5Cip6 (>2 µM). This suggests that the used concentrations (0.1 µM or 0.01 µM) for all three antibiotics or antibacterial compounds for TEM analysis will not have a similar effect on the bacterial cell. To observe similar antibacterial activity, the authors should use individual MIC values for each of the antibacterial compounds.

5.     Related to my previous comment, it is expected to see that the antibacterial activity of DL5Cip6 is lower than DL412 alone at both 0.01 and 0.1 µM concentrations for all three-time points (5, 15 and 45 min). If the long-term goal is to develop an effective hybrid antibiotic that can be used to treat infections, how this hybrid compound (DL5Cip6) works better than only an antibiotic (Cip) or synthetic compound (DL412)?

6.     Instead of showing several TEM images of a given time point, please show images of specific antibacterial compounds on a given culture at different time points (5, 15, 30 and 45 minutes). I suggest adding the images in panels for all antimicrobial compounds. It will be also helpful if the authors add the end result of the treatment, therefore images taken at 14-24 hours of incubation.

7.     The kinetic assay was done for 0,30,60,90,120,240,300 and 24 h culture on a 96-well plate and by absorbance reading. Table 1 shows data for 5, 15 and 45 min and cfu/ml suggesting the plate count was done. There is no mention of this experiment in the methods section. How this experiment was done?

8.     To inhibit the biofilm formation, a higher dose of antibiotics is needed. The authors used 0.1 µM and 0.5 µM of compounds to see the effect on biofilm formation. The authors should perform a separate assay to find out the minimal biofilm inhibitory concentration (MBIC) and use MBIC, 1/2MBIC and 1/4MBIC to see the effect of these compounds.

9.     Format the references according to the journal guideline.

10.  Italicize all organisms' name in the reference section.

11.  Please show the lengths of the scale for each TEM image.

12.  Please mention how many times a specific experiment was done. If the images shown were selected from many, please mention this or mention that the images shown here is a representative image. Also, mention, if you observed any differences in cell structure/damages in a given sample.

13.  Please mention how many times a particular experiment/assay was done.

14.  Please upload the supplementary file during the resubmission. I do not find any supplementary files.

Table 1: Please include the number of cells in scientific form.

Figure 2: Give the full name of ‘Cip’. What does it mean by ‘intact culture’. It may be written as ‘Representative images of intact S. aureus cells’. Show the lengths of the scale in the figure.

Figure 3: Are those figures from cultures taken at different time points?

Figure 5: Have you done this experiment for other time points (5 min, 15 min and overnight incubation)? How many times this experiment was done?

Line 14: Give the reference for the previous study.

Line 61: DL412 is not defined earlier. Provide the full name of the abbreviation when it appears first in the manuscript.

Line 94: Do you refer to the Cip, DL412, DL5Cip6 or all together by ‘their ability’?

Line 153: Does it mean that you did shaking of the culture at 580 rpm. If, so then its very vigorous shaking. Please specify whether it is shaking or centrifugation.

Line 174: Shaking at 180 rpm?

Line 181: If 5 min incubation leads to significant changes, then why did you choose 15 min incubation time for TEM?

Line 182: 40 or 45 min?

Moderate English correction is required for the manuscript.

Reviewer 2 Report

The manuscript by Grogoreva et al sheds light on the use of compounds that could have increased antibacterial activity, and also utilize TEM to study the morphological changes exerted by the compounds on the bacterial cell. The methodology executed seems to be technically sound and the results are presented in a clear and comprehensible manner. However, I have a few comments for the authors.

Major comments:

I would advise the authors to do an English grammar check on the text. There are a few instances where some passages are not clear. The authors can do it themselves.

The title should be changed to "Changes in the Ultrastructure of S. aureus Cells Make It Possible to Identify and Analyze the Antibacterial Effects of Hybrid 3 Compounds Composed of Ciprofloxacin and Polycationic Amphiphiles" as the authors didn't test a single "compound", but "compounds".

Minor comments:

Line 39 add a reference.

Line 143, add the full name of "LB".

Line 273 should be "the kinetics of the action of the compounds".

Line 288, in Table 1, DL5Cip6  caused a decrease in cell concentration even after 45 minutes, but in the text, the authors state that it "somewhat increased after 45 minutes". Kindly check if the statement is correct.

I think the authors should do a language check.

Round 2

Reviewer 1 Report

The revised version of the manuscript has improved, and the authors have addressed most of my comments, but not all. The main theme of the article is to visualize the antibacterial effect of ciprofloxacin, DL412 and DL5Cip6 on S. aureus cell structure and biofilm matrix by transmission electron microscopy (TEM). The authors found that the greatest damage occurred on the bacterial cell with DL412 and the effect of DL5Cip6 was not the combination effects of DL412 and ciprofloxacin, but rather more of the ciprofloxacin effect.

The manuscript can be significantly trimmed down by minimizing the texts in the results section. The revised version has separated the discussion from the results. However, I found the discussion is the repetition of the summary of the results instead of mentioning the significance of the work and compare with similar works (in other organisms or with other hybrid compounds) done previously. Please expand the discussion and trim the results section.

The authors also deleted the antimicrobial susceptibility of all three compounds from the previous version. I recommend adding the results to the manuscript and justifying the compound concentrations for TEM analysis and biofilm formation. The authors did not respond to my comments on using MIC, ½ MIC, or ¼ MIC concentration specific for each antimicrobial compound. 

About showing multiple TEM images at a given concentration and time, the authors responded that this has been shown in Figure S2. I appreciate that Figure S2 shows different concentrations. However, it does not show the effect of antibacterial compounds at different time points. Related to this, Figure S3 shows two incubation times (15 and 45 min), but with two different concentrations (0.1 µM and 0.01 µM, respectively) of DL412. I encourage the authors to show images for a given concentration at different time points for each of the compounds. If you have samples at 15, 30 and 45 min, then why do you prefer to show the images at 45 min incubation?

My comments about the selection of compound concentrations in biofilm assay have not been addressed properly. The authors used 0.1 µM and 0.5 µM to see the compounds' effect on biofilm. However, the authors should first identify the minimal biofilm inhibitory concentration (MBIC) and use MBIC, ½ MBIC, and ¼ MBIC in biofilm assay. The figure in the response file shows a concentration of up to 0.1 µM, but not with 0.5 µM, which was used to see the effect on biofilm. 

My specific comments are given below:

Line 19: Subscript ‘4’ in DL412. Similar in Line 23.

The last paragraph of the introduction needs to revise. The introduction should avoid the summary of the results rather than mentioning what this article is about.

Line 262: ‘M’ in µM should be upper case. Correct the format of the unit µM/ml.

Line 272: Is this statistically significant? Have you done any statistical analysis on the decrease in cell number? If not, then remove the word ‘significant’.

Line 278: I believe you meant log (CFU/ml).

Table 1: Give the full name of each of the antimicrobial compounds in the footnote.

Line 289: Add ‘is’ after the intermediate layer.

Figure 2: If possible, please show the length of the scale in the images (similar to Fig. 5) rather than in the figure legend. Same for Fig. 3 and 4.

Figure 5: What is the difference between Fig. 5 and Fig. S4 other than the different scale length? I found both figures have similar results.

Lines 742- 750 are more fitted in the discussion section.

Line 748-749: ‘All antimicrobial compounds used in this work caused pronounced and different changes in the structure of globules,’ is repeated twice.

Line 810: This should be Figure S4.

Figure S2 refers to the text prior to Figure S1. You may reorder the Supplementary figures.

Minor English correction is required.

Author Response

The response is attached. 
